# Do Bayesian Variational Autoencoders Know What They Don't Know?

**Misha Glazunov**[1]                          **Apostolis Zarras**[1]

[1]Delft University of Technology, the Netherlands

## Abstract

The problem of detecting the Out-of-Distribution (OoD) inputs is of paramount importance for Deep Neural Networks. It has been previously shown that even Deep Generative Models that allow estimating the density of the inputs may not be reliable and often tend to make over-confident predictions for OoDs, assigning to them a higher density than to the in-distribution data. This over-confidence in a single model can be potentially mitigated with Bayesian inference over the model parameters that take into account epistemic uncertainty. This paper investigates three approaches to Bayesian inference: stochastic gradient Markov chain Monte Carlo, Bayes by Backpropagation, and Stochastic Weight Averaging-Gaussian. The inference is implemented over the weights of the deep neural networks that parameterize the likelihood of the Variational Autoencoder. We empirically evaluate the approaches against several benchmarks that are often used for OoD detection: estimation of the marginal likelihood utilizing sampled model ensemble, typicality test, disagreement score, and Watanabe-Akaike Information Criterion. Finally, we introduce two simple scores that demonstrate the state-of-the-art performance.

## 1 INTRODUCTION

*Deep Neural Networks* (DNNs) are trained by *Maximizing the Likelihood Estimate* (MLE) over parameters $\boldsymbol{\theta}$ given the training input data $\mathcal{D}$: $p(\mathcal{D}|\boldsymbol{\theta})$. There exist two main approaches to modeling with DNNs: *discriminative* and *generative*.

The discriminative approach implies parameterizing a conditional distribution over target values $y$: $p(y|\mathbf{x}, \boldsymbol{\theta})$. The training of DNNs allows identifying optimum parameters

$\boldsymbol{\theta}^*$ based on a stochastic first-order optimization algorithm. In the case of classification tasks, the common choice for $p(y|\mathbf{x}, \boldsymbol{\theta})$ is a categorical distribution, in the case of regression—a Gaussian distribution (quite often with a constant variance). As it has been recently discovered: such models tend to be over-confident in their predictions with Out-of-Distribution (OoD) inputs [Nguyen et al., 2015, Hendrycks and Gimpel, 2016]. This discovery may not be surprising since MLE results in a point estimate and does not account for epistemic uncertainty. Taking into consideration the fact that in modern DNNs $|\boldsymbol{\theta}| \gg |\mathcal{D}|$, there may be several models $\boldsymbol{\theta}^*$ that generated $\mathcal{D}$.

Epistemic uncertainty can be estimated by inferring a posterior distribution: $p(\boldsymbol{\theta}|\mathcal{D})$ which can be done within the *Bayesian* frame of reference. Several promising results were achieved with the discriminative DNNs for OoD detection utilizing *Bayesian* inference over model parameters [Blundell et al., 2015, Chen et al., 2014, Maddox et al., 2019].

On the other hand, the generative approach allows learning the approximation of a true distribution over the training data: $p(\mathbf{x})$. DNNs again do the parameterization of this density, hence the name: *Deep Generative Models* (DGMs). Since DGMs provide a mechanism to estimate the probability of a particular input, they should supposedly assign a low density to the OoDs. However, recent research revealed that such estimations are prone to errors as DGMs often provide higher density values to OoDs than to *In-Distribution* (ID) data [Nalisnick et al., 2018].

As it was the case with the discriminative deep models, to overcome this problem, one may use a *Bayesian* DGM that infers the posterior: $p(\boldsymbol{\theta}|\mathcal{D})$ for the training data $\mathcal{D}$ over the model parameters $\boldsymbol{\theta}$. Such an approach allows getting an ensemble of the approximations of a true distribution of the data where each sample from the posterior $\boldsymbol{\theta} \sim p(\boldsymbol{\theta}|\mathcal{D})$ gives a separate instance of the model in the ensemble. Based on sampling from the posterior distribution, it is possible to estimate the density of the input instance $p(\mathbf{x})$ taking into consideration epistemic uncertainty.

*Accepted for the 38th Conference on Uncertainty in Artificial Intelligence* (UAI 2022).

In this work, we implement several methods that are widely applicable to the *Bayesian* inference over DNN parameters, namely: *Bayes by Backpropagation* (BBB) [Blundell et al., 2015], *Stochastic Gradient Hamiltonian Monte-Carlo* (SGHMC) [Chen et al., 2014], and *Stochastic Weight Averaging-Gaussian* (SWAG) [Maddox et al., 2019]. Most of the methods till now have been only applied to the discriminative supervised DNNs. It should be noted that even though the theoretical justification for *Bayesian Variational Autoencoders* (VAEs) was already present in the original paper [Kingma and Welling, 2013], there are still very few works addressing this point. In fact, to the best of our knowledge: there is only one paper dedicated to the BVAEs and OoD detection where only one of the methods (i.e., SGHMC) was used [Daxberger and Hernández-Lobato, 2019]. Our work represents an attempt to close this gap that is currently present between the discriminative and generative approaches based on DNNs. We transfer all of the mentioned methods to the deep generative VAEs and test them against several benchmarks suggested for OoD detection on various image datasets. Finally, based on our experiments, we introduce a couple of simple scores for the OoD detection that surpass all baseline scores.

In summary, we make the following main contributions:

- We perform the first implementation of three different *Bayesian* approaches for VAEs estimating epistemic uncertainty.

- We do a practical benchmarking of the most frequently used scores for OoD detection, taking into consideration *Bayesian* inference over the parameters of the likelihood of VAE.

- We suggest and apply two simple and efficient scores for the OoD detection that outperform baseline scores.

- We empirically evaluate the suggested approach based on several datasets.[1]

## 2 BACKGROUND

### 2.1 OUT-OF-DISTRIBUTION

Deploying a successful model requires the system to detect input data that are statistically anomalous or significantly different from those used during training. This is especially important for DNNs since they tend to produce overconfident predictions for such OoD inputs [Lee et al., 2018]. The lack of reliability of supervised discriminative models based on DNNs, when faced with OoD, was recently addressed by various methods [Hendrycks and Gimpel, 2016, Hendrycks et al., 2018, Liang et al., 2017].

---

[1] The source code for the reproducibility of the results is available at `https://github.com/DigitalDigger/BayesianVAEsOoD`

Unsupervised DGMs such as autoregressive models [Van den Oord et al., 2016], *Generative Adversarial Networks* (GANs) [Goodfellow et al., 2014], flow-based models [Dinh et al., 2016, Kingma et al., 2016, Kingma and Dhariwal, 2018], and VAEs [Kingma and Welling, 2013, Rezende et al., 2014] provide the opportunity to learn the density of the input data.

We choose to apply VAE as a particular instance of DGM in our experiments for several reasons:

1. It allows to obtain a particular value for the density in contrast to GANs that can only be sampled in a black-box manner.

2. It represents a model based on the latent variable, which seems like a reasonable assumption considering the complexity of the data's underlying density. The latent space has a much lower dimensionality in comparison with the dimensionality of the input. Such a bottleneck allows for learning the most relevant features. It distinguishes VAE from the flow-based models where all the transformations are invertible and represent a bijective mapping between the input and the latent space [Dinh et al., 2016, Kingma and Dhariwal, 2018, Nielsen et al., 2020].

3. It allows the parametrization of all the *Bayesian* inference constituents, including the posterior over latent variable and the likelihood of the data [Kingma and Welling, 2013, Rezende et al., 2014]. Such separation of the constituents makes it possible to work with the particular part, as in our case, with the decoder of the VAE for weight uncertainty estimation, distinguishing VAEs from the autoregressive models.

Furthermore, as it has been shown by Nalisnick et al. [2019], different DGM models may not produce similar results, which suggests that it is a good idea to concentrate only on one type for the analysis, which VAEs represent in our case.

However, it has been recently discovered that even in the case of DGMs that allows estimating the density, it does not work as intended and that DGMs return higher $p(\mathbf{x})$ for input data from a different distribution [Nalisnick et al., 2018].

There are several approaches to tackle this issue. One possible solution is to enhance the training dataset. Hendrycks et al. [2018] suggested incorporating the pre-selected anomalous examples employing the so-called outlier exposure technique, which achieved promising results. Ren et al. [2019] proposed a different method: two DGMs are trained separately—one for the semantics of the images and another for the background of the same images. The background images are generated with substantial noise to make the model learn this background, discarding the image's semantics. Then by calculating the likelihood ratios between the model that learned semantics and the model that learned

the background, it is possible to detect OoD. However, both of the suggested approaches rely on the knowledge of either the outlier or the image's background, which cannot be reasonably covered for all the possible inputs. Moreover, recent research reveals that the likelihood ratio method does not achieve satisfactory results when *a Bayesian* VAE is applied [Daxberger and Hernández-Lobato, 2019]. Due to these reasons, we do not consider methods of dataset enhancement in this work.

Another possible solution is to devise an alternative score for the OoD detection. In that vein, the *Watanabe-Akaike Information Criterion* (WAIC) was successfully used by Choi et al. [2018]. Further, the disagreement score [Daxberger and Hernández-Lobato, 2019] was suggested for the same purpose. This idea was motivated within the information-theoretic framework and was also based on the posterior estimation over the model parameters. The considered scores were calculated in both works based on the densities obtained from several models. In the former case, an ensemble was trained to calculate WAIC, while the latter case used the *Bayesian* VAE. In addition, Nalisnick et al. [2019] introduced a typicality test of the input sequences under the conjecture that the inlier sequences should be members of the DGM's typical set.

We chose to address the OoD problem similarly: i.e., we suggest and apply new simple scores that help to detect OoDs.

## 2.2 VARIATIONAL AUTOENCODER

VAE represents a type of DGM that provides the possibility of density estimation of the input $\mathbf{x}$. The optimization objective of VAE is the *Evidence Lower BOund* (ELBO), which allows joint optimization with respect to both variational parameters $\phi$ of the encoder responsible for the variational approximation of the posterior $q_\phi$ over the latent variable $\mathbf{z}$, and the generation parameters $\boldsymbol{\theta}$ of the decoder responsible for the parametrization of the likelihood of the input $p_\theta(\mathbf{x}|\mathbf{z})$:

$$\mathcal{L}_{\boldsymbol{\theta},\phi}(\mathbf{x}) = E_{q_\phi(\mathbf{z}|\mathbf{x})}[\log p_\theta(\mathbf{x}, \mathbf{z}) - \log q_\phi(\mathbf{z}|\mathbf{x})] \quad (1)$$

VAEs are trained in an unsupervised manner from data and are widely used for generative purposes.

## 2.3 ESTIMATION OF THE MARGINAL LIKELIHOOD

Marginal likelihood can be computed in the following way:

$$p_\theta(\mathbf{x}) = \int_{\mathbf{z}} p_\theta(\mathbf{x}|\mathbf{z})p(\mathbf{z})d\mathbf{z} \quad (2)$$

However, it is difficult to calculate it precisely due to the integration over the whole $\mathbf{z}$-space. As suggested by Rezende et al. [2014], as soon as the VAE is trained, it is possible to estimate the marginal likelihood of the input under the generative model using *importance sampling* w.r.t to the approximated posterior, namely:

$$p_\theta(\mathbf{x}) \simeq \frac{1}{N}\sum_{i=1}^{N} \frac{p_\theta(\mathbf{x}, \mathbf{z}_{(i)})}{q_\phi(\mathbf{z}_{(i)}|\mathbf{x})}, \quad \text{where} \quad \mathbf{z}_{(i)} \sim q_\phi(\mathbf{z}|\mathbf{x}) \quad (3)$$

As it has been discovered by Nalisnick et al. [2018], we cannot rely directly on the marginal likelihood estimations produced by a single DGM. This fact is not surprising for the discriminative models based on DNNs. Therefore, it should not be shocking for DGMs either, taking into consideration that they are also based on the DNNs and that they also obtain the optimal parameters $\boldsymbol{\theta}^*$ under the maximum likelihood estimation (MLE) for the $p(\mathcal{D}|\boldsymbol{\theta})$ which represents a point estimate. Hence, without *Bayesian* inference over model parameters, it is impossible to estimate the epistemic uncertainty, which results in the model's inability to provide a robust estimation of the marginal likelihood for OoD inputs.

## 2.4 EPISTEMIC UNCERTAINTY

The required posterior estimation over model parameters $p(\boldsymbol{\theta}|\mathcal{D})$ in the case of discriminative DNNs is usually implemented in the following three ways:

1. By using variational posterior approximation [Blundell et al., 2015];

2. By sampling from Markov Chain Monte Carlo [Chen et al., 2014];

3. By capturing the local geometry of the posterior through fitting a Gaussian with two first moments of the *Stochastic Gradient Descent* (SGD) [Maddox et al., 2019].

The first method represents a similar variational inference approach as in the case of VAEs when their encoders are trained to infer the posterior over the latent variable. The difference is that now it is applied to minimize the KL-divergence between the intractable posterior *over the model parameters* $p(\boldsymbol{\theta}|\mathcal{D})$ and the distribution from the family of tractable distributions $q(\boldsymbol{\theta}|\mathcal{D})$. It is also implemented by maximizing the ELBO. Since in our work we implement it in VAE, it means that we are making the variational inference for both the posterior for the latent variable conditioned on the input and the posterior for the parameters conditioned on the training data. Such an approach is called fully *Bayesian* in the case of VAEs [Kingma and Welling, 2013]. There are various methods for approximating the posterior for the model parameters; we will use the one suggested in [Blundell et al., 2015].

*Markov Chain Monte Carlo* (MCMC) constructs a Markov chain with the desired posterior as its equilibrium distribution. The most known method for MCMC is based on the

Metropolis-Hastings algorithm [Hastings, 1970, Metropolis et al., 1953]. This algorithm converges to the real posterior by exploiting the random walk proposal distribution. However, this convergence may be pretty slow due to the slow exploration of the state space based on the random walk. *Hamiltonian Monte Carlo* (HMC) [Duane et al., 1987] postulates the exploration within the framework of the Hamiltonian dynamics. It allows producing distant proposals for the Metropolis algorithm resulting in much faster convergence. We use a variant of HMC adopted for deep learning, namely SGHMC, that relies on the noisy gradient estimates [Chen et al., 2014].

Finally, it is possible to infer the desired posterior while training due to the noise in the SGD [Mandt et al., 2017]. We apply the *Stochastic Weight Averaging* (SWA) [Izmailov et al., 2018] together with fitting a Gaussian using the SWA solution as the first moment and covariance that is also derived from the SGD steps: the so-called SWAG method [Maddox et al., 2019]. SWAG is easy to implement since it does not require additional sampling and can be used as a baseline for the rest of the methods.

## 3 METHODOLOGY

### 3.1 BAYESIAN VAES

We implement several possible methods for *Bayesian* VAEs (BVAEs). We apply the *Bayesian* inference over the model parameters of the decoder of the VAE. Such a method allows sampling of several decoders to form an ensemble with the subsequent marginal log-likelihood estimation as indicated in Equation 3.

***Bayes by Backpropagation.*** We approximate the posterior distribution of the VAE decoder parameters given the training data $p(\boldsymbol{\theta}|\mathcal{D})$ based on the method suggested by Blundell et al. [2015]. This method was initially applied to discriminative learning. In our work, we implement it in VAEs. The ELBO objective is to find distribution parameters $\boldsymbol{\lambda}$ that minimize KL-divergence between our approximation and the true posterior; hence, the ELBO is formulated in the following way:

$$\mathcal{F}_{\boldsymbol{\theta}}(\mathcal{D}, \boldsymbol{\lambda}) = \mathbb{E}_{q(\boldsymbol{\theta}|\boldsymbol{\lambda})}\left[\log p(\mathcal{D}|\boldsymbol{\theta}) + \log p(\boldsymbol{\theta}) - \log q(\boldsymbol{\theta}|\boldsymbol{\lambda})\right] \quad (4)$$

$\log p(\mathcal{D}|\boldsymbol{\theta})$ represents the sum of the marginal likelihoods of the individual inputs:

$$\log p(\mathcal{D}|\boldsymbol{\theta}) = \sum_{i=1}^{N} \log p(\mathbf{x}^{(i)}|\boldsymbol{\theta}) \quad (5)$$

and

$$\log p(\mathbf{x}^{(i)}|\boldsymbol{\theta}) \geq \mathcal{L}_{\boldsymbol{\theta},\phi}(\mathbf{x}^{(i)}) \quad (6)$$

where $\mathcal{L}_{\boldsymbol{\theta},\phi}(\mathbf{x})$ is the ELBO for the marginal likelihood marginalized over the latent variable and it is defined in

Equation 1. Since ELBO is the lower bound of the marginal likelihood, we can use it for our approximation. The optimization objective is formulated as:

$$\widetilde{\mathcal{F}_{\boldsymbol{\theta}}}(\mathcal{D}, \boldsymbol{\lambda}) = \mathbb{E}_{q(\boldsymbol{\theta}|\boldsymbol{\lambda})}\left[\sum_{i=1}^{N}\left[\mathcal{L}_{\boldsymbol{\theta},\phi}(\mathbf{x}^{(i)})\right] + \log p(\boldsymbol{\theta}) - \log q(\boldsymbol{\theta}|\boldsymbol{\lambda})\right] \quad (7)$$

Now, if we plug in the right-hand side of the objective in Equation 1 we can get the Monte-Carlo estimation of the combined variational objective:

$$\widehat{\mathcal{F}_{\boldsymbol{\theta},\phi}}(\mathcal{D}, \boldsymbol{\lambda}) \simeq \frac{1}{L}\sum_{j=1}^{L}\left[\sum_{i=1}^{N}\left[\log p_{\theta^{(j)}}(\mathbf{x}^{(i)}, \mathbf{z}) - \log q_{\phi}(\mathbf{z}|\mathbf{x}^{(i)})\right] + \log p(\boldsymbol{\theta}^{(j)}) - \log q(\boldsymbol{\theta}^{(j)}|\boldsymbol{\lambda})\right] \quad (8)$$

where $\boldsymbol{\theta}^{(j)}$ is sampled from the posterior $q(\boldsymbol{\theta}|\boldsymbol{\lambda})$, $\boldsymbol{z}$ is sampled from the posterior $q(\boldsymbol{z}|\boldsymbol{x})$ and $N$ is taken equal to the batch size.

For the minimization objective, we use the negated estimate: $-\widehat{\mathcal{F}_{\boldsymbol{\theta},\phi}}(\mathcal{D}, \boldsymbol{\lambda})$. We assume a diagonal Gaussian distribution for both variational posteriors with parameters $\mu$ and $\sigma$. In order to make $\sigma$ be always non-negative we apply the same reparametrization as it was suggested by Blundell et al. [2015], namely $\sigma = \log(1 + \exp(\rho))$, yielding the following posterior parameters $\lambda = (\mu, \rho)$. For the prior over the latent variable, we use the standard normal density, for the prior over the weights we use the scale mixture of two Gaussians as in [Blundell et al., 2015].

The usual reparametrization trick [Kingma and Welling, 2013] is applied to both $\boldsymbol{\theta}$ and $\boldsymbol{z}$ for training by backpropagation.

***Stochastic Gradient Hamiltonian Monte Carlo.*** This approach exploits sampling instead of optimization, which was the case with BBB. This sampling is done within the MCMC framework and is based on the proposals generated utilizing the Hamiltonian dynamics. Namely, assume that the posterior distribution:

$$p(\boldsymbol{\theta}|\mathcal{D}) \propto \exp(-U(\boldsymbol{\theta}, \mathcal{D})) \quad (9)$$

where $U(\boldsymbol{\theta}, \mathcal{D})$ stands for the potential energy function in the Hamiltonian.

In our work, we take:

$$U(\boldsymbol{\theta}, \mathcal{D}) = -\log p(\boldsymbol{\theta}, \mathcal{D}) = -\sum_{i=1}^{N}\log p(\mathbf{x}^{(i)}|\boldsymbol{\theta}) - \log p(\boldsymbol{\theta}) \quad (10)$$

where $\log p(\mathbf{x}^{(i)}|\boldsymbol{\theta})$ is approximated by ELBO in our experiments and $\log p(\boldsymbol{\theta})$ is the prior over parameters.

Since HMC requires the computation of the gradient for the whole batch, a stochastic gradient alternative has been

suggested by Chen et al. [2014] which relies on the noisy gradients and allows proposal generation in a faster mini-batch manner.

In our work, we also apply the improvements suggested by Springenberg et al. [2016] which significantly reduce the number of hyperparameters through adaptive estimates of the parameters in question during the burn-in procedure and subsequent training. Such an approach has been previously implemented in the unsupervised generative setting with VAEs by Daxberger and Hernández-Lobato [2019].

***Stochastic Weighted Averaging-Gaussian.*** SWAG is fitting the following Gaussian distribution:

$$\mathcal{N}\left(\theta_{\text{SWA}}, \widehat{\Sigma_{\text{SWA}}}\right) \tag{11}$$

where $\theta_{\text{SWA}}$ is a running average over DNN parameters and $\Sigma_{\text{SWA}}$ is the sample covariance matrix that after $T$ epochs can be calculated as:

$$\theta_{\text{SWA}} = \frac{1}{T}\sum_{i=1}^{T}\theta_i \quad \text{and} \quad \Sigma_{\text{SWA}} = \frac{1}{T-1}\sum_{i=1}^{T}(\theta_i - \theta_{\text{SWA}})(\theta_i - \theta_{\text{SWA}})^\top \tag{12}$$

Since $\Sigma_{\text{SWA}}$ is of a very high rank it is approximated by the $K$ last epochs during training resulting in $\widehat{\Sigma_{\text{SWA}}}$[Maddox et al., 2019].

SWAG was previously applied only to the discriminative DNNs, in our work we implement it within a generative approach with VAEs.

***Combining several likelihoods.*** After the approximation of the variational posterior over the weights, the usual practice is to estimate the expected likelihood, the exact form of which can be formulated as follows:

$$p(\mathbf{x}^*|\mathcal{D}) = \int p(\mathbf{x}|\theta)p(\theta|\mathcal{D})d\theta \tag{13}$$

The unbiased estimate of which can be obtained like this:

$$\mathbb{E}_{p(\theta|\mathcal{D})}[p(\mathbf{x}^*|\theta)] \simeq \frac{1}{N}\sum_{i=1}^{N}p(\mathbf{x}|\theta_i); \quad \text{where} \quad \theta \sim p(\theta|\mathcal{D}) \tag{14}$$

$p(\mathbf{x}|\theta_i)$ is computed by importance sampling as in Equation 3. As soon as the expected likelihood is estimated, one can apply a threshold that would distinguish if the considered input adheres to the in-distribution sample or not.

In [Choi et al., 2018] the likelihoods returned by several generative models are used to estimate the WAIC:

$$WAIC(\mathbf{x}^*|\mathcal{D}) = \mathbb{E}_{p(\theta|\mathcal{D})}[p(\mathbf{x}|\theta)] - Var_{p(\theta|\mathcal{D})}[p(\mathbf{x}|\theta)] \tag{15}$$

WAIC estimates the gap between the expected likelihood and the variance between the obtained likelihoods, which should benefit the small variance cases.

Another alternative is calculating the disagreement score $D[\cdot]$ suggested by Daxberger and Hernández-Lobato [2019].

This score measures the variation in the likelihoods $\{p(\mathbf{x}^*|\boldsymbol{\theta}_i)_{i=1}^{N}\}$ which captures the uncertainty of the models within the ensemble about the particular input:

$$D_\Theta[\mathbf{x}^*] = \frac{1}{\sum_{\theta \in \Theta} w_\theta^2}; \quad \text{where} \quad w_\theta = \frac{p(\mathbf{x}^*|\boldsymbol{\theta})}{\sum_{\theta \in \Theta} p(\mathbf{x}^*|\boldsymbol{\theta})} \tag{16}$$

The lower the score, the more informative the input is about the parameters $\boldsymbol{\theta}$, and consequently, the uncertainty value is higher.

The weights represent the normalized likelihoods between 0 and 1. The disagreement score sums up the squares of the weights and takes the reciprocal. If the score is large, it means that all models return close values of the likelihoods. On the contrary, if the score is 1, then there is one model that dominates.

Finally, Nalisnick et al. [2019] conjectured that due to the high dimensionality of inputs, the over-confidence of DGMs may be due to the fact that in-distribution images lie in the typical set in contrast to the tested OoDs that concentrate in the high-density region. Based on this conjecture, they introduced the test for typicality that treats all input sequences of length $M$ as inliers if their entropy is sufficiently close to the entropy of the model, i.e., if the following holds for small $\epsilon$, then the given $M$-sequence is in-distribution:

$$\left|\frac{1}{M}\sum_{m=1}^{M} -\log p\left(\mathbf{x}_m; \boldsymbol{\theta}\right) - \mathbb{H}[p(\mathbf{x}; \boldsymbol{\theta})]\right| \leq \epsilon \tag{17}$$

We applied this score to one-element sequences since it is the most realistic scenario in practical applications of OoD detection.

***Our scores.*** Based on the results of the experiments with the available metrics on all of the considered methods, we decided to apply our scores that capture the variation. In our work, we apply two simple scores for the same purpose. First, we measure the information entropy of the normalized likelihoods, namely:

$$\mathbb{H}_\Theta[\mathbf{x}^*] = -\sum_{\theta \in \Theta} w_\theta \log w_\theta; \quad \text{where} \quad w_\theta = \frac{p(\mathbf{x}^*|\boldsymbol{\theta})}{\sum_{\theta \in \Theta} p(\mathbf{x}^*|\boldsymbol{\theta})} \tag{18}$$

It is a standard information-theoretic metric that measures the average information of the distribution: the lower the entropy, the more one of the models is confident about the predicted value. The entropy measure is applied to the normalized likelihoods. Such normalization may be considered as the categorical distribution over the obtained marginal likelihoods of the models.

Secondly, we calculate the sample standard deviation of the marginal log-likelihoods returned by the models within the ensemble:

$$\Sigma_\Theta[\mathbf{x}^*] = \sqrt{\frac{1}{N-1}\sum_{\theta \in \Theta}(\log p(\mathbf{x}^*|\boldsymbol{\theta}) - \overline{\log p(\mathbf{x}^*|\boldsymbol{\theta})})^2} \tag{19}$$

It measures the variation within the log-likelihoods directly without normalizing step as in the case of the entropy, so if the variation persists along with the considered methods and datasets, the standard deviation will capture this difference: the higher the value, the more uncertainty there is between the models about a particular input.

***Thresholding.*** There remains an open question of the appropriate threshold selection for the model evaluation. Since we are working in the unsupervised setting, intuitively, the ideal situation would be some threshold between the values of the scores returned by the model that successfully divides the inputs into OoDs vs. IDs. In order to validate the efficiency of the scores in achieving this task, we tackle this problem in the same way as Hendrycks and Gimpel [2016] by using three different metrics: *Area Under the Receiver Operating Characteristic Curve* (AUROC), the *Area Under Precision-Recall* (AUPR) curve, and the *False-Positive Rate at 80% of True-Positive Rate* (FPR80). These metrics are threshold-independent because they compute the true positives and false positives for all possible thresholds providing a final single value of the efficiency of the used non-thresholded decision values in dividing them into two separate classes.

## 4  EVALUATION

We run all of our experiments on the four image datasets: MNIST [LeCun and Cortes, 2010], Fashion-MNIST [Xiao et al., 2017], SVHN [Netzer et al., 2011], and CIFAR-10 [Krizhevsky et al., 2010]. As it has been observed [Ren et al., 2019, Nalisnick et al., 2018], the likelihood estimations may be misled by the dataset they have been trained on. For instance, if the model has been trained on MNIST and OoD detection has been performed on the Fashion-MNIST, then the researcher, only by chance, may obtain good results. To avoid such mistakes, we train our models on all the datasets and check the following in-distribution vs. OoD: MNIST vs. Fashion-MNIST and vice versa, SVHN vs. CIFAR-10 and vice versa.

The following hardware infrastructure was used in all of our experiments: Xeon Platinum 8160 2.1 GHz 32 GB of RAM, 1 GPU NVIDIA Volta V100.

First, we estimate the impact of the latent space's number of dimensions on the loss function. The dimensionality is closely connected with the dataset the model is trained on. MNIST and FashionMNIST results reveal no need to go over 10 latent dimensions since loss function did not significantly decrease after that value. For SVHN and CIFAR-10, we experimented with the number of latent dimensions up to 100; the most optimal results have been achieved with dimensionality equals 20 for SVHN and 70 for CIFAR-10.

We experimented with two different architectures for all our tests: one for the grayscale images and the second for the

RGB images with 1 and 3 channels correspondingly. All models have been trained for 1000 epochs. To evaluate the inputs, we sampled 200 different models for our ensemble and evaluated them on a separate test data split of 5120 images that models have not been trained on. We used test splits of both ID and OoD datasets for all scores and metrics.

For our implementation of Bayes by backpropagation, we noticed that random normal initializer of the DNNs weights suggested as a prior in the original paper by Blundell et al. [2015] resulted in very slow convergence. To speed up the process, we also experimented with the following parameters: random normal initializer with $0$ mean and $0.1$ standard deviations for $\mu$ and constant initializer for $\rho = -3$, which improved the training speed [Krishnan et al., 2020].[2]

In case of SGHMC we adhere to the same protocol as in Daxberger and Hernández-Lobato [2019], namely, we use the same scale-adapted sampler implementation with learning rate $10^{-3}$ for the training and momentum decay $0.05$.[3] We also place Gaussian priors over decoder parameters with precision $p(\theta) = \mathcal{N}(0, \lambda^{-1})$ and with Gamma hyperprior over the precision $p(\lambda) = \Gamma(\alpha, \beta)$ with $\alpha = \beta = 1$ that are resampled on each epoch.

For the experiments with SWAG, we set $K = 40$ and kept the default values for all the rest hyperparameters as in the original SWAG implementation.[4]

All the experiments are done within the framework implemented by Nielsen et al. [2020]. The results of the experiments against the benchmark scores for all datasets and models can be observed in the Tables 1 - 4.

## 5  DISCUSSION

As shown in the tables, *Bayesian* methods used in the discriminative approach can be successfully transferred to the generative models such as VAEs for OoD detection. Moreover, from the point of view of the scores: the variation among the model densities turns out to be persistent across all of the types of the *Bayesian* VAEs and all the datasets. The best results are achieved for BBB and SGHMC types of VAEs. The simple entropy score consistently demonstrates state-of-the-art results while detecting OoDs by better capturing the variation compared to the previously introduced baseline scores, the histograms for both BBB and SGHMC VAE results are shown in Figure 1. In addition, the sample standard deviation significantly outperforms the entropy score in the case of BBB and SGHMC methods.

---

[2]For BBB we used the PyTorch *Bayesian* layers available at `https://github.com/IntelLabs/bayesian-torch`

[3]The SGHMC sampler that we used is available at `https://github.com/automl/pybnn`

[4]SWAG sampler that we used is available at `https://github.com/wjmaddox/swa_gaussian`

Table 1: Scoring values across all types of *Bayesian* VAEs trained on Fashion-MNIST data and tested on MNIST as OoD

| | Fashion-MNIST vs. MNIST | | | | | | | | |
| | BBB | | | SGHMC | | | SWAG | | |
| | ROC AUC↑ | AUPRC↑ | FPR80↓ | ROC AUC↑ | AUPRC↑ | FPR80↓ | ROC AUC↑ | AUPRC↑ | FPR80↓ |
|---|---|---|---|---|---|---|---|---|---|
| **Expected LL** | 40.43 | 45.46 | 95.20 | 40.43 | 45.18 | 94.99 | 25.09 | 38.01 | 99.54 |
| **WAIC** | 59.53 | 59.35 | 71.88 | 55.79 | 53.86 | 74.56 | 19.90 | 35.14 | 99.83 |
| **Typicality test** | 40.51 | 43.40 | 86.36 | 41.02 | 43.85 | 86.05 | 56.40 | 50.32 | 64.88 |
| **Disagreement score** | 96.44 | 97.22 | 1.11 | 95.25 | 96.31 | 2.50 | 79.98 | 80.74 | 38.24 |
| **Entropy (ours)** | 97.97 | 98.43 | **0.19** | 97.28 | 97.92 | **0.53** | **82.50** | **84.05** | **35.61** |
| **Stds of LLs (ours)** | **99.64** | **99.55** | 0.34 | **99.56** | **99.50** | 0.55 | 19.90 | 34.22 | 94.42 |

Table 2: Scoring values across all types of *Bayesian* VAEs trained on CIFAR-10 data and tested on SVHN as OoD

| | CIFAR-10 vs. SVHN | | | | | | | | |
| | BBB | | | SGHMC | | | SWAG | | |
| | ROC AUC↑ | AUPRC↑ | FPR80↓ | ROC AUC↑ | AUPRC↑ | FPR80↓ | ROC AUC↑ | AUPRC↑ | FPR80↓ |
|---|---|---|---|---|---|---|---|---|---|
| **Expected LL** | 59.73 | 53.27 | 58.99 | 60.39 | 53.74 | 58.08 | 60.31 | 53.51 | 57.05 |
| **WAIC** | 61.15 | 54.22 | 57.15 | 62.39 | 55.38 | 55.07 | 64.29 | 55.81 | 50.59 |
| **Typicality test** | 63.73 | 60.89 | 65.53 | 64.44 | 61.05 | 64.01 | 64.93 | 61.52 | 64.33 |
| **Disagreement score** | 81.16 | 84.82 | 38.47 | 80.41 | 83.00 | 40.61 | 73.27 | 76.95 | 54.95 |
| **Entropy (ours)** | 84.76 | **88.21** | 29.31 | 84.56 | 86.90 | 29.12 | **76.51** | **80.54** | 49.37 |
| **Stds of LLs (ours)** | **89.98** | 85.83 | **16.03** | **92.52** | **91.48** | **12.27** | 71.26 | 64.65 | **44.34** |

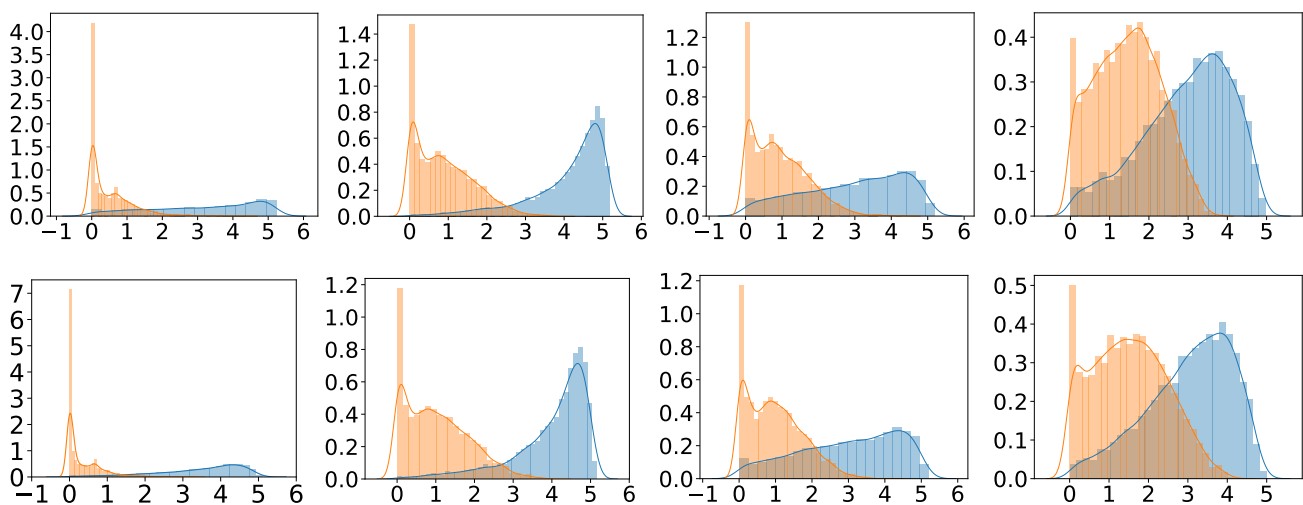

Figure 1: Histograms of the entropies of the marginal likelihoods. LLs are estimated based on sampling from the *Bayesian* VAEs, blue depicts in-distribution (ID) and orange - out-of-distribution (OoD). **From left to right**: MNIST as ID vs Fashion-MNIST as OoD, Fashion-MNIST as ID vs MNIST as OoD, SVHN as ID vs CIFAR-10 as Ood, CIFAR-10 as ID vs SVHN as OoD. **Top:** Sampling is done from Bayes-by-backprop VAE. **Bottom:** Sampling is done from SGHMC VAE.

Interestingly, almost all scores achieve comparably good results when trained on the MNIST dataset and tested on Fashion-MNIST (Table 3), but many of the baseline scores demonstrate substantially worse values when the experiments are conducted the other way around, i.e., trained on Fashion-MNIST and tested on MNIST (Table 1). The reason is that many of these scores are biased to a particular type of data. However, the bi-directional experiments easily identify these biases among all datasets and benchmark scores.

It can be observed that the worst-performing scores either intrinsically depend on the mean of the ensemble (such as WAIC) or on the log-likelihood itself returned by the model (such as a typicality score). In the first case, it results in the dominance of the variation of the particular values of the likelihoods for different inputs over the variation between the models within the ensemble for a single input, e.g., the range of variance of estimated likelihoods in the case of Fashion-MNIST is at least twice greater than in

Table 3: Scoring values across all types of *Bayesian* VAEs trained on MNIST data and tested on Fashion-MNIST as OoD

| | *MNIST vs. Fashion-MNIST* | | | | | | | | |
| | BBB | | | SGHMC | | | SWAG | | |
| | ROC AUC↑ | AUPRC↑ | FPR80↓ | ROC AUC↑ | AUPRC↑ | FPR80↓ | ROC AUC↑ | AUPRC↑ | FPR80↓ |
|---|---|---|---|---|---|---|---|---|---|
| **Expected LL** | 99.98 | 99.98 | 0.00 | 99.93 | 99.92 | 0.04 | **96.83** | **96.20** | **5.18** |
| **WAIC** | 99.99 | 99.99 | 0.00 | **99.94** | **99.94** | 0.02 | 80.37 | 76.25 | 33.56 |
| **Typicality test** | 99.98 | 99.98 | 0.00 | 99.88 | 99.90 | 0.00 | 94.91 | 96.47 | 1.58 |
| **Disagreement score** | 98.95 | 99.01 | 0.23 | 97.32 | 97.70 | 1.37 | 94.88 | 93.97 | 8.99 |
| **Entropy (ours)** | 99.42 | 99.47 | 0.02 | 98.50 | 98.75 | 0.29 | 95.72 | 95.20 | 8.37 |
| **Stds of LLs (ours)** | **99.99** | **99.99** | **0.00*** | 99.91 | 99.91 | **0.00*** | 80.37 | 82.78 | 39.12 |

\* 0's are possible since it is a value for false-positive rate at 80% of true-positive rate

Table 4: Scoring values across all types of *Bayesian* VAEs trained on SVHN data and tested on CIFAR-10 as OoD

| | *SVHN vs. CIFAR-10* | | | | | | | | |
| | BBB | | | SGHMC | | | SWAG | | |
| | ROC AUC↑ | AUPRC↑ | FPR80↓ | ROC AUC↑ | AUPRC↑ | FPR80↓ | ROC AUC↑ | AUPRC↑ | FPR80↓ |
|---|---|---|---|---|---|---|---|---|---|
| **Expected LL** | 58.65 | 61.79 | 77.72 | 57.09 | 60.56 | 80.18 | 58.98 | 62.06 | 76.52 |
| **WAIC** | 64.46 | 66.01 | 68.39 | 62.17 | 64.38 | 72.45 | 62.84 | 68.42 | 75.25 |
| **Typicality test** | 44.63 | 44.28 | 81.46 | 43.35 | 43.63 | 82.45 | 44.28 | 44.13 | 81.96 |
| **Disagreement score** | 85.20 | 88.35 | 30.26 | 85.31 | 88.52 | 28.66 | 77.58 | 80.36 | 45.60 |
| **Entropy (ours)** | 87.80 | 90.63 | 20.77 | 87.89 | 90.76 | 19.91 | **80.01** | **83.24** | **41.58** |
| **Stds of LLs (ours)** | **93.29** | **91.51** | **10.99** | **94.70** | **93.95** | **8.67** | 59.31 | 53.36 | 61.78 |

case of MNIST. In general, the more complex dataset is used for model training, the greater variance of the resulting likelihoods that the model assigns to the inputs; hence, there is potentially less influence of the variation between the models within the ensemble (that is completely lost in the case of WAIC for example). On the other hand, our scores measure the variance *within* the ensemble. It allows catching even a slight difference in such variation. In the second case with typicality, the log-likelihoods of inputs are used directly without any ensembling, which is susceptible to the well-known issue with modern deep generative models discovered by Nalisnick et al. [2018]. The same applies to the expected log-likelihood metric.

From the point of view of the speed performance, we consider the runtime required for the training convergence to get the same values of the likelihoods as for the vanilla VAEs. In such a case, the overhead of SWAG is almost negligible compared with the vanilla VAE. BBB and SGHMC, on the contrary, both take much longer time, with BBB requiring up to five times longer than the vanilla VAE training. SGHMC performs relatively faster than BBB but still lags far behind SWAG. There is also a clear tradeoff between the training performance and resulting accuracy in distinguishing between OoD vs. ID inputs: the fastest method in training (i.e., SWAG) results in the lowest OoD detection scores; however, the much slower training method (i.e., SGHMC) results in the best OoD detection scores for the most complex dataset that we experimented with (i.e., CIFAR-10). BBB turns out to be the slowest in training, and the results for OoDs are slightly worse than those obtained by SGHMC. Since

SWAG has a minor overhead during training, it implies better scalability of this method to bigger datasets compared with SGHMC or BBB. If we also consider the speed performance from the point of view of the scalability to the bigger models, then it becomes clear that all of the methods rely on sampling the weights, so there is no clear winner, i.e., the more parameters a particular DNN would have the slower sampling would be for all of the methods.

It should be emphasized that these discrepancies in speed performance between the suggested *Bayesian* approaches can be seemingly treated as limitations in comparison with the vanilla VAEs and with the different ensemble techniques. First, let us consider the time required for the convergence of the model. As mentioned above, it may take five times longer in the worst case to obtain the values comparable to the ones of the vanilla VAE for both components of the ELBO loss. However, this disadvantage is disappearing, and the proposed *Bayesian* methods become even advantageous when one is training an ensemble of separate models $\{p(\mathbf{x}^*|\boldsymbol{\theta}_i)\}_{i=1}^N$ with $N > 5$ under the similar hardware constraints. Second, suppose we look at the resulting detection of OoD, which is slowed down with the several estimations of the marginal likelihood within the ensemble and subsequent calculation of the required score. In that case, this limitation primarily concerns the performance comparison with a single DGM without using ensembles. However, as we mentioned before, such a point estimate cannot be reliably used to estimate epistemic uncertainty. Therefore, if we compare with the traditional ensembling techniques, the main difference with the suggested approach stems from

the way *how* the estimated marginal likelihoods are used. The former computes the expected likelihood, and the latter calculates the entropy or sample standard deviation of the marginal log-likelihoods, which means that we get $\mathcal{O}(N)$ computations in either of these scenarios.

# 6 CONCLUSION

The ability to detect OoD inputs by DGMs is of significant importance for robust inference, especially in practical applications. Our work concentrates on a specific type of such DGMs: Bayesian VAEs. We addressed this issue from three different perspectives:

1. ***Method-wise.*** We implemented three methods for estimating epistemic uncertainty in the generative setting based on VAEs utilizing *Bayesian* inference over model parameters: BBB based on variational inference, SGHMC based on Monte-Carlo sampling, and SWAG based on the noise in the SGD. Most methods have been previously applied exclusively to the discriminative models, and our paper bridges this gap between two modeling approaches.

2. ***Score-wise.*** We benchmarked all methods against the frequently used OoD benchmarks: expected log-likelihood, WAIC, disagreement score, and typicality test. Moreover, during our experiments, we noticed that the most promising score was based on the idea of the variation of marginal likelihoods. Built on that, we proposed using two simple scores: one is based on the information entropy, and the second is on the standard deviation for the robust unsupervised OoD detection. We achieved state-of-the-art results with them across all the benchmarked methods and considered datasets.

3. ***Experiment-wise.*** We did thorough experiments with all methods and scores on several datasets. Moreover, to avoid potential errors, we evaluated the results bi-directionally, e.g., if we trained a model on the MNIST dataset and used Fashion-MNIST as OoD, then we also trained a model on the Fashion-MNIST dataset and checked its ability to detect MNIST inputs as OoD. Such a check is necessary to avoid the bias of any particular scoring method to either more complicated or more simplified data.

The results of the experiments convincingly support the idea of the beneficial usage of the epistemic uncertainty estimation based on the variation for successful OoD detection in the case of VAEs. We observed that both BBB and SGHMC demonstrated comparable performance. While SWAG was always worse for the new OoD scores compared with other methods, we still conclude that it can be used as a simple baseline for epistemic uncertainty in the case of VAEs in the same manner as in the case of the discriminative approach. Moreover, from the point of view of the training

convergence, SWAG turned out to be the fastest among the all considered *Bayesian* methods. Future work may revolve around a deeper understanding of the sources of variation within the ensemble from the point of view of the latent space of the VAE: e.g., is there a correlation between "holes" in the latent manifold and greater variance of the likelihoods.

### Acknowledgements

This project has received funding from the European Union's Horizon 2020 research and innovation programme under grant agreement No 883275 (HEIR).

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
