# OpenReview forum: "Do Bayesian Variational Autoencoders Know What They Don't Know?"
_auai.org/UAI/2022/Conference — UAI 2022 Poster_

### Official Review · Reviewer_eHpT · 2022-03-18

**Q2(1) Originality/Novelty:** 2
**Q2(2) Significance/Impact:** 2
**Q2(3) Correctness/Technical Quality:** 4
**Q2(6) Clarity Of Writing:** 1
**Q6 Overall Score:** 5
**Q8 Confidence In Your Score:** 4

**Q1 Summary And Contributions:**

The paper attempts to empirically study whether the epistemic uncertainty (in the form of OODs) can be captured by Bayesian neural networks (BNN).

However, I feel uncertain about the main contribution of the paper. Looking at the four main contributions listed on page 2, I feel that all contributions are mechanical and do not reveal any new message for machine learning. See my comments below


------
After the authors' explanation, I decided to raise the score to borderline acceptance.

**Q2 Assessment Of The Paper:**

More detailed information regarding each of these aspects is given below:

**Q2(4) Quality Of Experiments (Optional):**

4: Excellent: The experimental evaluation is comprehensive and the results are compelling.

**Q2(5) Reproducibility:**

4: Excellent: Key resources (e.g., proofs, code, data) are available and key details (e.g., proof sketches, experimental setup) are comprehensively described for competent researchers to confidently and easily reproduce the main results.

**Q3 Main Strengths:**

The paper is experimentally solid. The choice of methodology is clearly explained, the evaluation is thorough and the experimental details are sufficient.

**Q4 Main Weakness:**

Maybe this is a problem with the paper writing, I cannot find any explicit important machine learning message in the current draft. This problem has two manifestations. Let me explain.

1. The question asked by the title is not explicitly answered: "Do Bayesian variational autoencoders know what they don't know?" After reading through the paper, I do not find an explicit answer to or related discussion about this. I feel that the paper is either mistitled or the paper writing is not good enough (maybe under-discussed or misstructured)

2. The paper has minimal (almost nonexisting) text that discusses the actual finding. I find 99% of the paper discussing implementation details and only 1% stating and discussing the actual finding. Let me be more specific in the next section

**Q5 Detailed Comments To The Authors:**

In my opinion, the findings are too under-discussed.

The only few sentences that state and discuss the finding are on page 7, which I quote here: "The results of the experiments for all datasets and models can be observed in the Tables 1 - 4. As it can be seen, Bayesian methods used in the discriminative approach can be successfully applied also for the generative models such as VAEs for OoD detection. Moreover, it can be observed that the variation among the model densities persists across all of the types of the Bayesian VAEs and all the datasets. The best results are achieved for BBB and SGHMC types of VAEs."

There is no discussion of how this finding relates to the question raised in the title. There is no discussion of its practical or theoretical implications. There is no discussion about how the finding improves our understanding of deep learning or Bayesian learning.

I feel that the authors need to improve this part significantly. Perhaps the authors should spend a separate and extended section discussing the meaning of the finding.

**Q7 Justification For Your Score:**

Unclear and insufficient discussion of the results. See the above comemnts.

**Q9 Complying With Reviewing Instructions:**

1: Yes.

---

### Official Review · Reviewer_zXqH · 2022-04-03

**Q2(1) Originality/Novelty:** 2
**Q2(2) Significance/Impact:** 2
**Q2(3) Correctness/Technical Quality:** 3
**Q2(6) Clarity Of Writing:** 2
**Q6 Overall Score:** 5
**Q8 Confidence In Your Score:** 4

**Q1 Summary And Contributions:**

The paper considers the use of Bayesian neural networks for out-of-distribution detection in the context of VAEs. The aim is to compare the different sampling/approximation techniques of the weight posterior and their impact on OoD detection quality. The authors also propose their own metrics for OoD that perform relatively well in experiments, including MNISt/FMNIST and SVHN/CIFAR-10.

**Q2 Assessment Of The Paper:**

More detailed information regarding each of these aspects is given below:

**Q2(4) Quality Of Experiments (Optional):**

3: Good: The experimental evaluation is adequate, and the results convincingly support the main claims.

**Q2(5) Reproducibility:**

2: Fair: Key resources (e.g., proofs, code, data) are unavailable but key details (e.g., proof sketches, experimental setup) are sufficiently well-described for an expert to confidently reproduce the main results.

**Q3 Main Strengths:**

Very interesting analysis of Bayesian neural networks for VAEs and the relationship to OoD. In principle, the paper is very helpful to gain a better understanding of how Bayesian approaches compare in that regard.

The presentation of the different approaches is very thorough and clear and (almost, see below) all experimental details are outline. Hence, the paper is easy to follow for the most part.

**Q4 Main Weakness:**

There are some minor flaws in the presentation, writing and referencing. For details, see below. Apart from that, the main issue that I could see was that with all the experimental description, there was very little said about the actual metrics of the experiment. The experiment section provides very little analysis of the figures and tables so their content and logic needs to be inferred from context. Similarly, the experiments themselves aren't really defined. What exactly is classified in Tables 1-4? I assume it is a classification of the OoD vs ID data points, but I could not find that clearly written anywhere. Did you take the test set for both datasets? How many data points? More details are definitely needed including equations.

Please let me know if I missed this, but I read through several times.

The experiment section could be longer, including e.g. Langevin sampler and more datasets to draw strong conclusions.

**Q5 Detailed Comments To The Authors:**

You say
> When the considered Bayesian epistemic uncertainty method requires a density function, we substitute it
by its approximation, i.e., by ELBO.

I don't understand this. In the previous sentence you say you can approximate the likelihood using importance sampling? What exactly are you trying to approximate?

"Normal" -> "normal"

You cite Metropolis and Ulam (1949) for the Metropolis Hastings, but the correct paper is
> Metropolis, N., Rosenbluth, A. W., Rosenbluth, M. N., Teller, A. H., & Teller, E. (1953). Equation of state calculations by fast computing machines. The journal of chemical physics, 21(6), 1087-1092.

Similarly, Hamiltonian Monte Carlo (originally called Hybrid Monte Carlo) is originally due to
> Duane, Simon; Kennedy, Anthony D.; Pendleton, Brian J.; Roweth, Duncan (3 September 1987). "Hybrid Monte Carlo". Physics Letters B. 195 (2): 216–222


**Q7 Justification For Your Score:**

I am on the fence on this one. Nice topic and provides a useful discussion that I would like to see at the conference. The experiment section could contain more detail, but I think I would be okay with the current selection of algorithms and datasets. The problem at the moment is my confusion as to the exact experimental settings and explanations of the numbers without which I think the paper might be difficult to understand.

After discussion: see below.

**Q9 Complying With Reviewing Instructions:**

1: Yes.

---

### Official Review · Reviewer_V5ej · 2022-04-11

**Q2(1) Originality/Novelty:** 2
**Q2(2) Significance/Impact:** 2
**Q2(3) Correctness/Technical Quality:** 3
**Q2(6) Clarity Of Writing:** 3
**Q6 Overall Score:** 6
**Q8 Confidence In Your Score:** 3

**Q1 Summary And Contributions:**

The authors focus on one significant concern in deep learning, the detection of out-of-distribution OoD on variational auto-encoders VAE, a class of deep generative models. They implement several Bayesian approaches to track the uncertainty of the inference under the model parameter space. The authors then proceed to evaluate comparing with baselines and present the positive results of their entropy and standard deviation of likelihoods approaches.

**Q2 Assessment Of The Paper:**

More detailed information regarding each of these aspects is given below:

**Q2(4) Quality Of Experiments (Optional):**

3: Good: The experimental evaluation is adequate, and the results convincingly support the main claims.

**Q2(5) Reproducibility:**

2: Fair: Key resources (e.g., proofs, code, data) are unavailable but key details (e.g., proof sketches, experimental setup) are sufficiently well-described for an expert to confidently reproduce the main results.

**Q3 Main Strengths:**

The paper presents a good comparison of different approaches to estimate the uncertainty of the inference while accounting for the uncertainty of the model parameters.
The authors used different implementations for the different approaches, yet the resulting graphs as shown in Fig. 1, indicate very similar results, which adds credibility to the comparison.
The comparison of the different metrics show potential future avenues of research, as they indicate what seems to work and nor, for OoD detection.

**Q4 Main Weakness:**

The only major concern I have, is that the values in the AUC for the CIFAR-SVHN seem to be too good, given the overlap from Fig. 1.


**Q5 Detailed Comments To The Authors:**

The paper focuses on an area of general interest for practitioners. Measures of awareness of whether the models are fit for the data being used are of significant importance for robust inference.

Here the authors focus on the evaluation of different metrics while introducing two more which show promising results.
Furthermore, they also shoe empirically that the Bayes by Backpropagation (BBB) approach provides similar results to the Stochastic Gradient Hamiltonian Monte Carlo (SGHMC), which helps the reader confirm that the approach up to sampling was working correctly.

In general I like the paper, it brings several approaches together and introduces a couple of extra measures. I believe the paper would be stronger bringing more discussion into your new metrics. Maybe more theoretical developments on how your Stds of LLs compare to WAIC, as they seem very close up to a scaling factor and the square root.

Although there is a significant amount of effort spent in the paper, as training and evaluating such models is time consuming. The novelty is really on the new metrics and they are a very small fraction of the whole paper.

One minor suggestion is to align the scales on Fig. 1, label them and better describe the axis. This would help the reader when comparing the graphs.

**Q7 Justification For Your Score:**

Although the authors have spent significant effort training and evaluating the models under different settings. I'm concerned with the values in table 2. and Fig.1. In particular, the case of CIFAR vs SVHN where the entropy presents a very significant overlap yet the AUC is still quite high. Maybe more details on the computation for the AUC for the entropy would be beneficial.

**Q9 Complying With Reviewing Instructions:**

1: Yes.

---

### Official Review · Reviewer_1qwr · 2022-04-15

**Q2(1) Originality/Novelty:** 3
**Q2(2) Significance/Impact:** 2
**Q2(3) Correctness/Technical Quality:** 3
**Q2(6) Clarity Of Writing:** 3
**Q6 Overall Score:** 7
**Q8 Confidence In Your Score:** 3

**Q1 Summary And Contributions:**

This paper focuses on answering the question of whether bayesian inference of parameters of vae's can improve OoD capabilites by having
a better estimated of the epistemic uncertainity.

They applied 3 different Bayesian infernece approaches on vaes and emperically compared the approches over common metrics used for OoD detection.

They introduced a new score based on information entropy of normalized likelihoods, and showed their new score performs much better for detecting OoD.


**Q2 Assessment Of The Paper:**

More detailed information regarding each of these aspects is given below:

**Q2(4) Quality Of Experiments (Optional):**

3: Good: The experimental evaluation is adequate, and the results convincingly support the main claims.

**Q2(5) Reproducibility:**

3: Good: Key resources (e.g., proofs, code, data) are available and key details (e.g., proofs, experimental setup) are sufficiently well-described for competent researchers to confidently reproduce the main results.

**Q3 Main Strengths:**

- Overall, the paper was well written, and easy to follow. The background section and related work also also decent.
- The paper also attemps to solve an important problem of detecting OoD using generative models, and specifially focuses on whether adding baysian inference on top of VAEs can help the models to account of epistemic uncertainity.
- The new score seems to be decent and better at detecing OoD.
- The expereimnts are well described and comprehensive

- See Q5 for the rest.

**Q4 Main Weakness:**

- Although not the main focus of paper, I think adding runtime and mentioning compute infastructure used for the experiments are always a good addition.
There was a small discussion about runtime in appendix but more detail would have been nice. For example, I would be curious to know how much do we give up in terms of speed performance when doing baysian inference (for each approach and each scoreing method).

- Some quick discussion about advantages or disadvantages of each of the 3 methods (BBB, SGHMC, SWAG) would have been nice. For example, on things like which ones scale better
on bigger datasets or bigger models.

**Q5 Detailed Comments To The Authors:**

- The new provided score i.e. based on entropy of the normalized likelihood also seems to perform much better than other common score used when tyring to detect OoD examples
using generative models.

- There is comprehensive experiments with all the different measure combined with multiple pair of datasets, overall the experiments seem adequate
to showcase their new proposed score does better in detecting OoD using bayesian vaes compared to other common scores.

- Is there an intution why all methods seem to do great (">99+") in many cases  on table 3 (trained on mnist tested on fashion), but much worst the other way around in table 1 (trained on fasion tested on mnist)?

- For FPR80, it says value for false-positives rate at 80% of true positive rates. Are they rates computed from training datasets or from the test datasets? Basiaclly we need to compute i) a threshold that gives 80% true positive rate (is that on trian or test) ii) use that threshold to compute false-positive (also is this computed on test or train)?

**Q7 Justification For Your Score:**

The main theoreical contribution seems to be the new score based on entropy to detect OoDs.

Emperical contributions are the comprehensive comparison of different bayesan inference methods on VAEs on multiple pair of datasets.



**Q9 Complying With Reviewing Instructions:**

1: Yes.

---

### Decision · Program_Chairs · 2022-05-15

**Decision:**

Accept (Poster)

**Comment:**

Meta Review: The paper considers the use of Bayesian neural networks for out-of-distribution detection in the context of VAEs. The goal is to compare the different sampling/approximation techniques of the weight posterior and their impact on OoD detection quality. It also provides a good comparison of different approaches to estimate the uncertainty of the inference while accounting for the uncertainty of the model parameters.

Rebuttal: the authors have carefully replied to reviewers' comment